# The conservation value of human-modified landscapes for the world's primates

Carmen Galán-Acedo[1], Víctor Arroyo-Rodríguez [1], Ellen Andresen[1], Luis Verde Arregoitia [2], Ernesto Vega[1], Carlos A. Peres[3] & Robert M. Ewers[4]

Land-use change pushes biodiversity into human-modified landscapes, where native ecosystems are surrounded by anthropic land covers (ALCs). Yet, the ability of species to use these emerging covers remains poorly understood. We quantified the use of ALCs by primates worldwide, and analyzed species' attributes that predict such use. Most species use secondary forests and tree plantations, while only few use human settlements. ALCs are used for foraging by at least 86 species with an important conservation outcome: those that tolerate heavily modified ALCs are 26% more likely to have stable or increasing populations than the global average for all primates. There is no phylogenetic signal in ALCs use. Compared to all primates on Earth, species using ALCs are less often threatened with extinction, but more often diurnal, medium or large-bodied, not strictly arboreal, and habitat generalists. These findings provide valuable quantitative information for improving management practices for primate conservation worldwide.

[1] Instituto de Investigaciones en Ecosistemas y Sustentabilidad, Universidad Nacional Autónoma de México, Antigua Carretera a Pátzcuaro no. 8701. Ex-Hacienda de San José de la Huerta, 58190 Morelia, Michoacán, Mexico. [2] Instituto de Ciencias Ambientales y Evolutivas, Universidad Austral de Chile, Campus Isla Teja, 5090000 Valdivia, Chile. [3] Centre for Ecology, Evolution and Conservation, School of Environmental Sciences, University of East Anglia, Norwich, Norfolk NR4 7TJ, UK. [4] Dept of Life Sciences, Imperial College London, Silwood Park Campus, Buckhurst Road, Ascot, Berkshire SL5 7PY, UK. Correspondence and requests for materials should be addressed to C.G.-A. (email: cgalanac@gmail.com) or to V.A.-R. (email: victorarroyo_rodriguez@hotmail.com)

With ~70% of all terrestrial ecosystems currently altered by human activities[1], the preservation of biodiversity and ecosystem functions is challenging[2], particularly in the tropics[3]. As a consequence of land-use change, an increasing number of species are being "forced" to inhabit human-modified landscapes, which are constituted by a mosaic of different land covers, both natural and anthropic. The ability of organisms to use anthropic land covers (ALCs) is rapidly becoming a key determinant of their persistence in human-modified landscapes[3,4]. Therefore, a better understanding of how and why some species use different types of ALCs is urgently needed to better predict and manage biodiversity in the Anthropocene[5].

Traditionally, the ALCs surrounding remnants of natural vegetation are referred to as the matrix[4,6]. However, rather than being regarded as a homogeneous land cover of unsuitable habitat, as assumed in early classical models[7,8], the anthropic matrix should be viewed as a collection of different ALCs, many of which can be used by species for different purposes, including foraging, dispersal, and reproduction[6,9]. In fact, there is evidence for birds, frogs, small mammals, and ants showing that the higher the ability of a species to use ALCs, the lower their probability of becoming extinct in fragmented landscapes[9–11]. In other words, patch-dependent species typically have higher extinction thresholds[12], meaning they require larger amounts of unmodified habitat to avoid extinction[13]. Unfortunately, for many species, studies mostly focus on their ecology within their primary habitat, especially in protected areas[14], thus limiting our understanding of their use of and tolerance to ALCs. This information is urgently needed to shed light on many theoretical debates about the main drivers of biodiversity patterns in human-modified landscapes.

The predominance of the habitat–matrix paradigm (i.e., binary landscapes comprised of either habitat or nonhabitat) in landscape ecology has been strongly criticized[15,16] and is gradually being replaced by approaches based on heterogeneous landscapes[17,18]. Emerging ecological approaches, such as "countryside biogeography"[19] and different theoretical models[20,21] and debates (e.g., land-sharing vs. land-sparing debate[22,23]) are based on the premise that the matrix is in fact heterogeneous, and that each ALC type may span a spectrum of species-specific ecological value. To better understand species' responses to landscape changes we need to assess the ecological role of each land cover (e.g., provision of food, refuge, and nesting sites) to be able to design functional landscapes[18]. This information can be used to improve management and conservation strategies. For instance, if species are relatively resilient to changes in their habitat and able to use resources in ALCs, they will fare better with a land-sharing approach that limits land-use intensification at the potential cost of increased habitat conversion[24]. Alternatively, if species are highly sensitive to habitat changes and are unable to use ALCs, a land-sparing approach will be more effective as it maximizes natural habitat conservation while concentrating production elsewhere[22].

Nonhuman primates (primates, hereafter) are particularly susceptible to land-use changes[25], which threaten ~60% ($n = 278$ species) of the world's 504 species with extinction[26]. As most primate species are forest-specialists, particularly in the Neotropics[27] forest loss is considered a main threat to primate conservation[28]. There are, though, many local, and landscape characteristics that may help reduce the impact of habitat loss on primate survival in human-modified landscapes[29]. However, most research has focused on assessing the effects of the characteristics of natural vegetation remnants on primate diet, behavior, and demography[30]. While primates are known to use some types of ALCs[31–33], the available evidence is widely scattered and the global patterns of use remain unknown beyond a qualitative

level. Further, no comprehensive effort exists to link primates' ecological traits to the extent of use of specific ALCs, greatly limiting our ability to predict the impact of specific landscape-management strategies on these mammals.

Here, we provide quantitative evidence regarding which types of ALCs are most frequently used by primates and for what activities. We also evaluate whether there are certain characteristics of the species, such as conservation status, ecological traits, and/or phylogenetic relationships that may help us predict their use of ALCs. We address these questions by reviewing 468 records of ALC use by primates. We focus on the most common ALC types in human-modified landscapes, including human settlements, open areas (i.e., annual crops and cattle pastures), tree plantations, connectors (i.e., isolated trees and linear landscape elements such as live fences and hedgerows), and secondary forests (i.e., regenerating forests following the removal of native vegetation). We compare the characteristics of species using these ALCs with the expected values based on all of the world's primates. The primate characteristics considered were conservation status (IUCN conservation category and population trends), ecological traits (diel activity, locomotion, trophic guild, body mass, and forest specialization), and phylogenetic relationships.

## Results and Discussion

**Use of anthropic land covers by primates.** We found positive evidence that at least 147 primate species (~30% of 504 primate species on Earth) use at least one of the five ALC types, with 60 genera (out of 82 genera in the world, ~75%) and all 15 families represented. Use of ALCs was evident worldwide (Fig. 1a), but the percentage of species was significantly higher than expected by chance in mainland Africa, and lower than expected in Madagascar (chi-squared test, $\chi^2 = 15.78$, $P = 0.001$; Fig. 1b). Different ALC types varied in the number of species using them ($\chi^2 = 20.64$, $P < 0.001$; Fig. 1c): secondary vegetation was used by the highest number of species (79) and human settlements were used by the lowest (34). This is not surprising, as these two types of ALC represent two extremes in a gradient of habitat modification. This pattern was particularly evident in the Neotropics and in Madagascar (Fig. 1d), where most species are strictly arboreal. On the other hand, a higher proportion of primates from mainland Africa were recorded using human settlements and open areas and a higher proportion of primates from Asia used tree plantations, human settlements, and open areas such as annual crops and cattle pastures (Fig. 1d). This is probably because many primate species in these two biogeographic realms have both arboreal and terrestrial locomotion modes. In some regions of these realms this pattern can also be caused by peoples' perception of primates as sacred animals, which favors their persistence in human-dominated environments[34,35].

All ALC types were used for foraging, resting, and traveling (Fig. 2). Human settlements and secondary forests were mostly used for either foraging or all activities combined, suggesting that these ALCs can be used as temporary or permanent habitats under certain conditions. Although most studies did not report if the species were using ALCs as habitat, at least 86 species (17% of all primates on Earth) are actively obtaining food resources from ALCs, highlighting their importance for primate conservation[32,36]. In the case of forest-specialist primates, which represent 70% of the studied species, these results suggest that they can supplement their diet by foraging in ALCs—a process referred to as landscape supplementation[20]. Connectors, such as living fences and isolated trees, supported primate foraging for 24 species, but almost half of the records were for travel alone, demonstrating the important role of these ALCs in increasing landscape

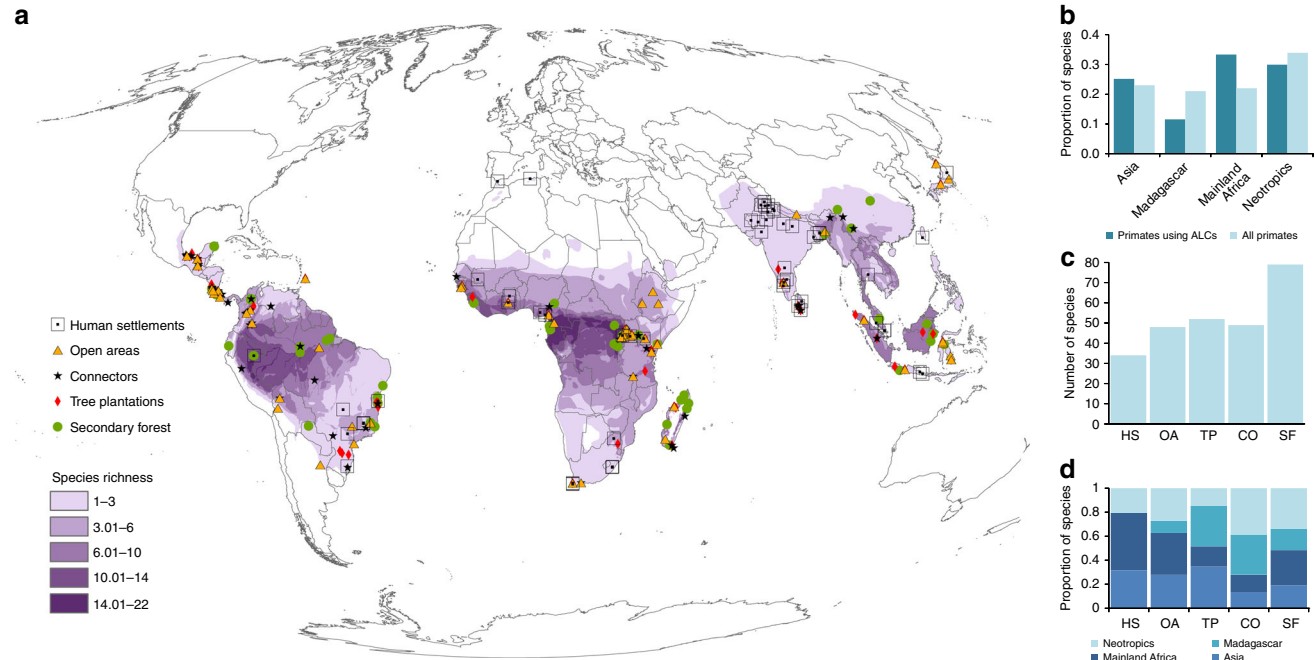

**Fig. 1** Geographic distribution of studies reporting the use of different anthropic land covers (ALCs) by primates. Spatial location of each study and global pattern of primate species richness (**a**). Proportion of species using ALCs ($n = 147$ species) compared to the total proportion of species ($n = 504$ species) in each biogeographic realm (**b**). African primates were classified in two groups, those from mainland Africa and those from Madagascar, because these two land masses span the distribution of two highly divergent primate suborders (catarrhines and strepsirrhines, respectively). Number of primate species recorded using each of five ALCs (**c**). Proportion of primate species using each ALC type in each realm (**d**). ALCs are categorized as human settlements (HS), open areas (OA), tree plantations (TP), connectors (CO), and secondary forests (SF). Species richness data in (A) was extracted from Pimm et al.[58]

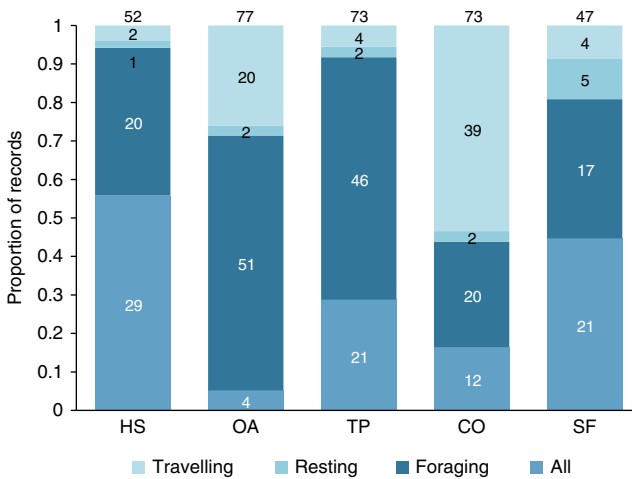

**Fig. 2** Activities of primates in each anthropic land cover (ALC) type. The proportion (and total numbers above each column and within column sections) of records for traveling, resting, foraging or all activities in different ALC types is indicated. ALCs, include human settlements (HS), open areas (OA), tree plantations (TP), connectors (CO), and secondary forests (SF). The total number of records varies because one study can report activities for more than one primate species whereas others do not report any primate activity

connectivity[32,37,38]. An important next step will be to assess which species can maintain their populations solely in ALCs, which species are strongly dependent on their natural habitats, and which ones may survive in natural habitat patches with some degree of landscape supplementation in ALCs.

**Conservation signal in the use of anthropic covers**. We found a significant, positive relationship between the use of ALCs and both conservation status and population trend (IUCN red list[39]; IUCN 2017 Fig. 3). The proportion of species classified as least concern was significantly higher in the group of primates recorded using ALCs, compared to all primates, particularly in human settlements ($\chi^2 = 18.95$, $P < 0.001$). Nearly half of all species recorded using ALCs were classified as vulnerable, endangered, or critically endangered by the IUCN (Fig. 3a), suggesting that ALC use alone does not necessarily prevent endangerment. Although use of ALCs may favor primate persistence in human-modified landscapes, it is important to recognize that their use also exposes primates to important threats, such as hunting, road kills, predation, and infectious diseases[40–42].

About 80% of all species using ALCs showed declining population sizes (Fig. 3b). Nonetheless, ALC use seems to soften this pattern, as we found a lower proportion of species with decreasing populations using ALCs than would be expected based on the world's primates. The latter pattern was particularly strong for primates using human settlements ($\chi^2 = 25.52$, $P < 0.001$) and open areas ($\chi^2 = 10.67$, $P = 0.005$). These results suggest that species able to use highly modified ALC types have a higher probability of persisting in anthropogenic tropical landscapes.

**Ecological traits that predict the use of anthropic covers**. We found significant associations between the ecological traits of primates and their use of ALCs (Fig. 4). In particular, nocturnality was less frequent among species using ALCs, especially in open areas ($\chi^2 = 13.88$, $P < 0.001$), secondary forest ($\chi^2 = 11.58$, $P = 0.003$), connectors ($\chi^2 = 9.62$, $P = 0.008$), and human settlements ($\chi^2 = 8.52$, $P = 0.014$; Fig. 4a). We would have expected a higher (not lower) incidence of nocturnality among species using ALCs because nocturnal primates are less likely to

   **3**

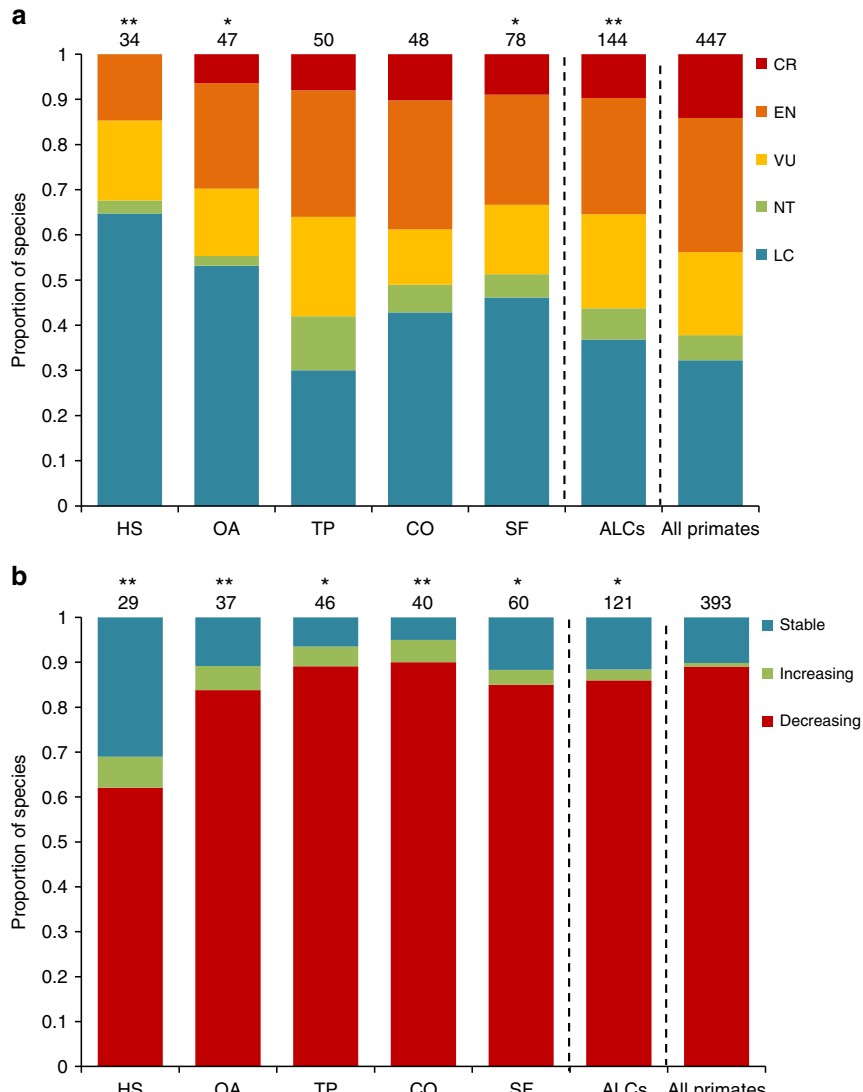

**Fig. 3** Conservation status and population trends of primate species using anthropic land covers (ALCs) compared to all the world's primates. The proportion (and total number above bars) of primate species within each IUCN threat category (**a**), and population trend (**b**), are shown. We tested for differences in frequencies with chi-square tests of goodness of fit ($^{*}P \leq 0.05$, $^{**}P \leq 0.01$) between primates using ALCs and all the world's primates, by separately assessing primate species that used different types of ALCs (HS human settlements, OA open areas, TP tree plantations, CO connectors, SF secondary forest). From higher to lower extinction risk, threat categories include: critically endangered (CR), endangered (EN), vulnerable (VU), near threatened (NT), and least concern (LC). We excluded species classified as data deficient and not evaluated in (**a**), and those whose population trends are unknown in (**b**)

encounter humans, and thus, they could perceive ALCs as less dangerous, compared to diurnal primates[43]. Yet, nocturnal primates are all forest-specialists, arboreal and with small-to-medium body mass—ecological traits that together seem to limit the use of ALCs (see below).

Strictly arboreal species were less frequent in the group of primates using ALCs, particularly among those using human settlements ($\chi^2 = 19.66$, $P < 0.001$) and open areas ($\chi^2 = 13.95$, $P < 0.001$; Fig. 4b). Similarly, there was also a lower proportion of small-bodied species using ALCs than expected by chance, particularly, once again, among those using human settlements ($\chi^2 = 12.43$, $P = 0.002$) and open areas ($\chi^2 = 19.01$, $P < 0.001$; Fig. 4c). The latter result is not surprising as small-bodied species are more likely to be arboreal, which limits their movement into treeless areas. Also, small primates tend to have smaller home ranges[44], and thus, they can be able to inhabit smaller habitat remnants without using resources from ALCs. This may lower the

probability of observing them in ALCs, especially in landscapes with a relatively recent history of anthropic land use (e.g., <30 y). Although this finding does not mean that small primates have lower extinction risk in human-modified landscapes[45], our results point in this direction, as 57.3% of large ALC-tolerant species ($n$ = 21 species) are threatened with extinction, whereas 49.9% of small-bodied species ($n = 42$) are threatened (Supplemntary Table 1). In contrast, other studies suggest that those species that avoid using ALCs can be more prone to extinction in these emerging landscapes;[10,46,47] thus, additional primate studies are needed to accurately assess the effect of body weight on extinction risk.

Forest-specialists were present in all land cover types, but they were less frequent among ALC-tolerant species ($\chi^2 = 11.19$, $P = 0.003$), particularly in human settlements ($\chi^2 = 31.53$, $P < 0.001$) and open areas ($\chi^2 = 11.76$, $P = 0.003$) (Fig. 4d). Although primate trophic guild was not significantly related to the use of

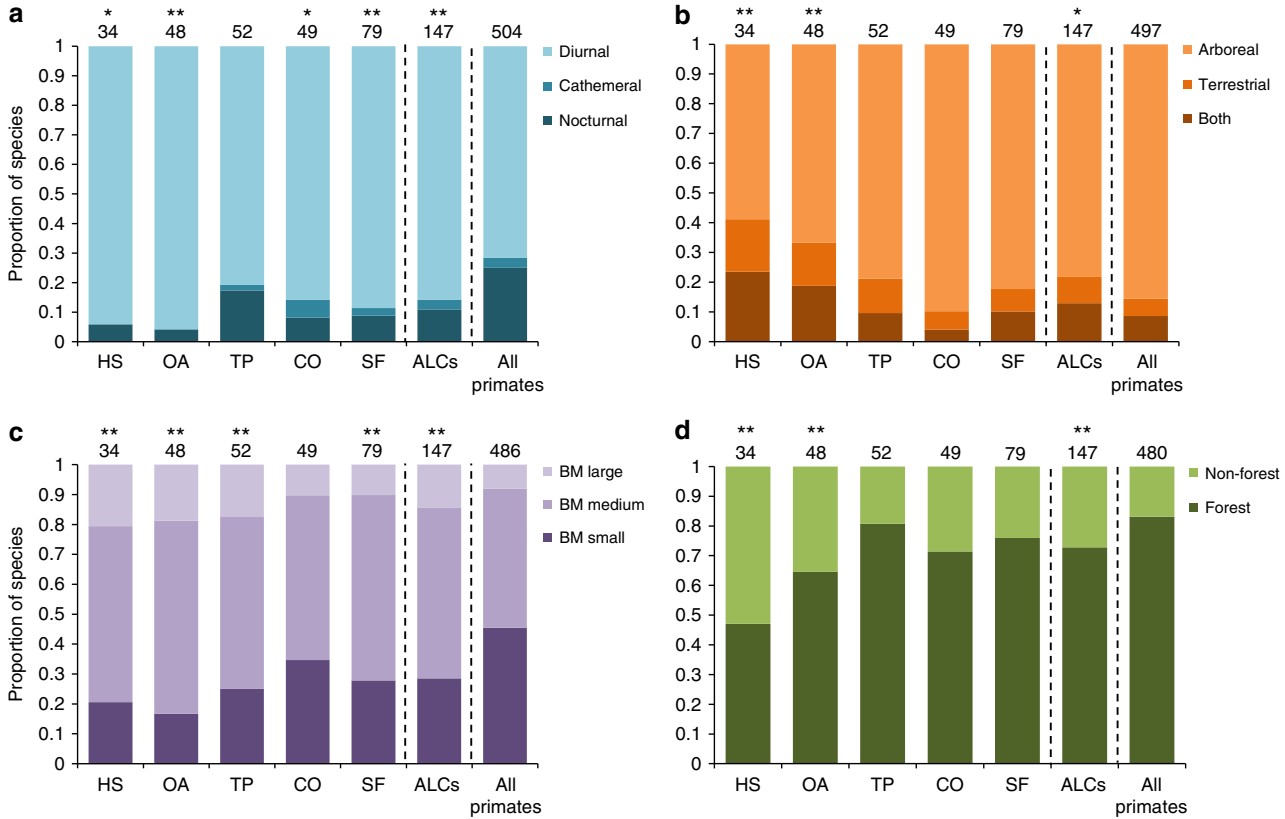

**Fig. 4** Ecological traits of primate species that have been recorded using anthropic land covers (ALCs). The proportions (and total number above bars) of primate species exhibiting different diel activity patterns (**a**), modes of locomotion (**b**), body mass classes (**c**), and forest specialization or not (**d**). We tested for differences in frequencies with chi-square tests of goodness of fit ($^{*}P \leq 0.05$, $^{**}P \leq 0.01$) between primates using ALCs and all the world's primates, by separately assessing primate species that used different ALC types (HS human settlements, OA open areas, TP tree plantations, CO connectors, SF secondary forest). Body mass (BM) was classified as small (<2 kg), medium (2–10 kg), or large (>10 kg). We excluded from analyses those species for which we found no information

ALCs, there was a trend toward a higher proportion of omnivorous species in human settlements than expected (Supplementary Fig. 1). These results, together with the fact that strictly arboreal species were less frequent in ALCs, suggest that the more generalist a species is, especially in terms of habitat and/or locomotion, the more resilient it is to habitat disturbance; a finding consistent with previous studies[46–48].

**Phylogenetic signal in ALC use.** We found a very weak phylogenetic signal in the use of ALCs (Fig. 5), where it was neither clustered nor randomly distributed across the phylogenetic tree ($D = 0.83$; $P [D = 0] < 0.001$; $P [D = 1] = 0.001$). The sensitivity analyses revealed that removing Cercopithecidae, the primate family with the largest number of species analyzed, did not influence the estimates of phylogenetic signal. However, the removal of sportive lemurs (family Lepilemuridae) significantly influenced our estimates of phylogenetic signal, despite having a similar number of species to most other families (see Supplementary Note 1). In particular, our results indicate that most species in this primate family (Lepilemuridae) do not use ALCs, i.e., nonuse of ALCs is a phylogenetically conserved characteristic for this clade. The highly conserved morphology and shared ecological traits (e.g., arboreal locomotion, nocturnal activity, and forest specialization) within this family[49] could explain this pattern. In contrast, the behavioral and ecological traits that could make a species tolerant to ALC conditions vary in their degree of phylogenetic conservatism[50]. This, together with the widespread

alteration of primates' habitats, leads to a pattern in which the use of ALCs is unpredictable in relation to species' evolutionary relationships.

**Conclusions.** Given the ongoing loss and alteration of primates' natural habitats, knowledge about how and why some species are able to use ALCs is essential to propose effective conservation strategies in human-modified landscapes. We provide a comprehensive quantification of the use of five dominant types of ALCs by primates worldwide. We also provide a global assessment of the relationships between primate use of ALCs and primates' ecological traits, conservation status and phylogenetic relationships. Our findings highlight the fact that ALCs can play important roles for the conservation of many primate species in anthropogenic landscapes, providing food resources, refuge and opportunities for dispersal. We note, however, that for 70% of the primates on Earth, we found no evidence of ALC use, suggesting that benefits associated with the use of ALCs are limited to some species, in which case they are unlikely to prevent the current extinction crisis of the world's primates[26]. While some poorly studied species might also be able to exploit ALCs, many other species are likely to depend on remnants of their primary habitat for their long-term conservation. Also, the use of ALCs can have negative effects on primates' populations, as it increases both exposure to several threats and occurrence of conflicts with humans due to crop raiding, aggression, or disease transmission[40–42]. Thus, although priority conservation actions should

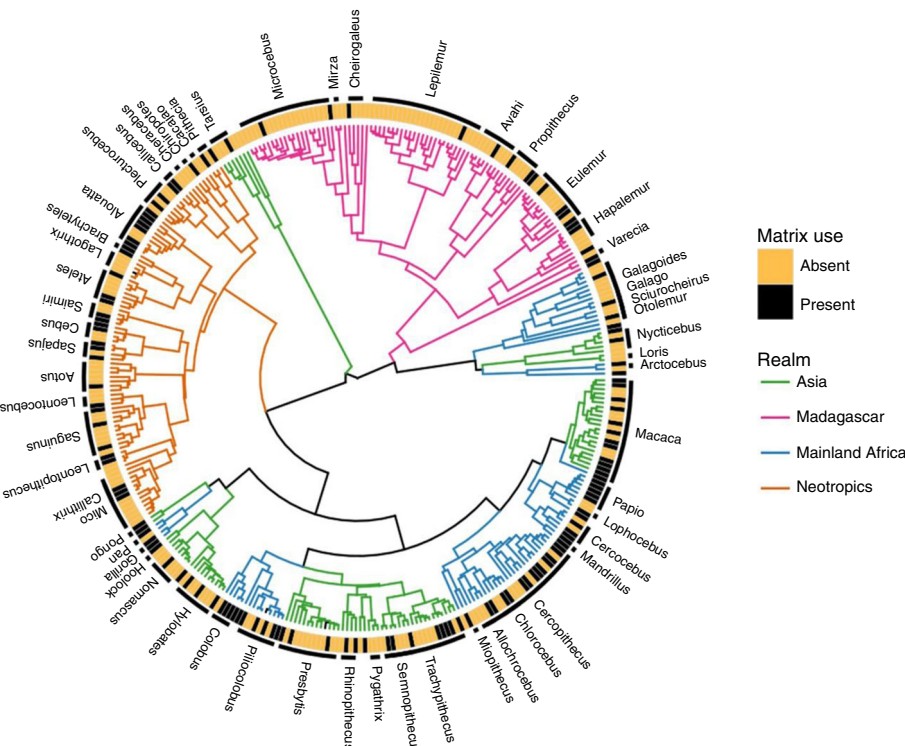

**Fig. 5** Distribution of the use of anthropic land covers (ALCs) across the primate phylogeny. Use of ALCs (present/absent) is indicated for each of the 352 species with phylogenetic data, based on the molecular timetree of Dos Reis et al.[50]. All genera with ≥2 species are labeled, and branches are color-coded by realm

focus on the maintenance of primary habitats for primates and other vertebrate taxa[9], they can be complemented with other land-management strategies, such as replacing highly modified ALCs by more functional land covers that provide resources for wildlife and/or facilitate their movement between habitat patches. Such an integrative approach will enhance the conservation value of increasingly modified landscapes for our closest relatives.

## Methods

**Evidence of ALC use by primates**. We systematically searched for articles published up to November 2, 2016, using the following search term sequence in ISI Web of Knowledge (www.isiwebofknowledge.com), SciVerse SCOPUS (www.scopus.com) and Google Scholar (https:// scholar.google.com.br/) databases: [(primate* OR monk*) AND ("plantation" OR "crop" OR "agroecosystem" OR "cultivation" OR "agriculture" OR "regenerating forest" OR "regenerating vegetation" OR "secondary forest" OR "secondary vegetation" OR "second growth" OR "clear cut" OR "life fence" OR "isolated trees" OR "scattered trees" OR "remnant trees" OR "corridor" OR "fencerow" OR "corridor line" OR "bridge" OR "stepping stones" OR "fence" OR "connectivity" OR "hedgerow" OR "strip" OR "city" OR "urban" OR "human settlement" OR "village" OR "settlement" OR "pasture" OR "grazing line" OR "ground" OR "cattle" OR "ground") AND/OR ("fragmentation" OR "landscape")]. These keywords were searched across all reference topics, except in Web of Knowledge where searches were restricted to title, abstract and keywords of articles. We then conducted additional searches in Google Scholar using keywords translated into Portuguese, Spanish and French, including the grey literature (e.g., MSc and PhD theses and unpublished reports). We classified all hits obtained into five groups, depending on the type of ALC used by primates: (i) human settlements (i.e., any kind of urban environment such as cities, towns, or villages), (ii) open areas (i.e., annual crops and cattle pastures), (iii) tree plantations (including all types of agroforestry systems), (iv) connectors (i.e., isolated trees and linear landscape elements, such as vegetation corridors, live fences and hedgerows), and (v) secondary forests (i.e., regenerating forests following regrowth after an acute disturbance event, such as logging and deforestation). We excluded review articles and studies with captive or reintroduced animals. Because for some ALCs the available literature is scarce, we selected the most recent 60 studies per each ALC type. Nevertheless, as some studies included information about more than one ALC type, the final database included 258 independent studies (Supplementary Table 1) containing 468 records of 147 primate species using ALCs. Such records

span 44 countries from four biogeographic realms: mainland Africa (17 countries), Madagascar, Asia (13 countries), and the Neotropics (13 countries).

From each study, we obtained, for each primate species, the scientific name and family, geographic coordinates and country, and the activity recorded within the ALC (i.e., traveling, resting, foraging, or all activities). Traveling refers to movements within and between ALC types. Resting refers to short/long diurnal/nocturnal rests, and foraging refers to the procurement, acquisition and/or ingestion of food. We assumed that resting and foraging require travel to reach any given destination. Therefore, the category of "all activities" included studies that reported observations on all three main activities, or resting and foraging, in an ALC. As most studies did not report extended information about the use of each ALC, we cannot know if primates are using it as habitat. Information about the surrounding landscape, such as distance to the nearest edge, proportion of remaining primary habitat, were not reported in the vast majority of studies, thereby precluding analyses related to these types of information. Furthermore, although studies reported general coordinates of the study sites, most did not report the coordinates where the individuals were recorded in an ALC, limiting our capacity to assess the landscape context.

**Conservation and ecological predictors**. For each primate species we obtained the conservation status, the population trend, and whether they are forest-specialists or not, from the International Union for Conservation of Nature (IUCN) database in the "letsR"[52] package for R, version 3.0.1[53]. Regarding ecological traits, we considered: locomotion mode (i.e., arboreal, terrestrial or both), diel activity (i.e., diurnal, nocturnal, or cathemeral), and body mass and trophic guild. Although body mass is a morphological trait, we considered it as an ecological trait because of its very high-ecological significance. For instance, it is positively related to home range size, thus affecting the way species interact with their environment[44,54]. Body mass was categorized into three classes: small (<2 kg), medium (2–10 kg), and large (>10 kg). Trophic guilds included six general groups: frugivorous (>60% of fruits in diet), folivorous (>60% leaves in diet), folivorous–frugivorous (diet comprised of both fruits and leaves in similar proportions), omnivorous (both plants and animals in diet), insectivorous (diet dominated by arthropods) and gummivorous (diet dominated by plant exudates). Ecological trait data was primarily extracted from Mittermeier et al.[27]. When some of the ecological traits were not reported in this encyclopedia, we actively searched for information in the literature. When the trait was reported in other scientific articles or databases, we searched for 1–3 references (depending on availability) and we used mean or modal values[55]. In total, we reviewed 370 studies, most of them published in peer-reviewed scientific journals and books. For each specific

datum we included the corresponding reference. The database was carefully checked for possible errors. When a specific datum was considered nonreliable (e.g., very extreme or contradictory values and values obtained with questionable methodology) we did not include it in the database. To assess relationships between primate characteristics and ALC use, we used goodness of fit chi-square tests. We selected this analysis because it is particularly recommended to compare observed vs. expected frequencies. In particular, we compared the number of species with each trait between those species that were observed using ALCs and the expected values based on all of the world's primates. We excluded from analyses species for which there was no available information.

**Phylogenetic signal**. To quantify phylogenetic signal in ALCs use, we used published phylogenetic relationships and divergence times from a molecular timetree built using 79 gene segments for 372 species (367 primates and 5 outgroup species) and 8 fossil-calibrated nodes[50, 51]. Specifically, we used the timetree built considering autocorrelated rates of molecular evolution (identified by Bayesian model selection as fitting the data better than a model with independent rates), and a conservative interpretation of both the age and the placement of key fossils with the living primate radiation. Of the 367 species included in this phylogeny, we retained 352 after standardizing synonyms and dropping infraspecific taxa.

To explore how phylogeny might capture species differences in ALCs use, we calculated the $D$ statistic[56]. $D$ measures phylogenetic signal strength in binary traits. Values of $D$ are scaled to set points of 0.0 (trait values phylogenetically conserved as expected under a Brownian Motion threshold model) and 1.0 (trait values distributed randomly across the phylogeny). For significance testing, the observed distribution of trait values at the tips of the tree was compared to both randomly shuffled values and the expected values from a Brownian Motion threshold model. For all tests of phylogenetic signal, we used the phylo.d function in the R package "caper"[57]. We used 9999 permutations to estimate the probability of the observed value of $D$ under null models of both no phylogenetic structure and Brownian motion.

To investigate the nonrandom but weak result we found for the Order as a whole, we tested whether certain clades were driving relatively large changes in our estimates of phylogenetic signal. We adapted the framework provided in the "sensiPhy"[59] package to perform sensitivity analyses and tested how excluding families (with ten or more species) from the analyses would influence the estimates of phylogenetic signal. When the removal of a clade leads to a large change in the estimate of $D$, it can be considered to be influential. To correct for clade size, we used randomization tests to determine if the change in parameter estimates is significantly different from a null distribution created by randomly removing the same number of species as the focal clade (Supplementary Table 2, Supplementary Fig. 2).

**Reporting summary**. Further information on experimental design is available in the Nature Research Reporting Summary linked to this article.

## Data availability

All data generated or analyzed during this study are included in this published article or its Supplementary Information Files.

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

## Acknowledgments

CONACyT (Grant 2015-253946) provided financial support. C.G.-A. received a scholarship from CONACyT (Mexico) and a grant from Rufford (18689-1). We gratefully acknowledge P.A.D. Dias for his valuable and constructive suggestions on the manuscript. We are also grateful for the support provided by the Instituto de Investigaciones en Ecosistemas y Sustentabilidad, UNAM. Part of the writing was done while V.A.-R. was on sabbatical at Carleton University, funded by PASPA-DGAPA-UNAM, and C.G.-A. was on a research visit at the Imperial College London, UK.

## Author contributions

C.G.-A. and V.A.-R. designed the research project, with advice from E.A. and L.V.A. C.G.-A. reviewed literature and collected the data. C.G.-A., L.V.A. and V.A.-R. analyzed data. C.G.-A., V.A.-R. and E.A. led the writing of the manuscript and E.V., C.A.P. and R. M.E. offered significant feedback on the manuscript.

## Additional information

**Competing interests:** The authors declare no competing interests.

