## [Peer Review File · Nature Communications]

Reviewers' comments:

Reviewer #1 (Remarks to the Author):

Nature comms review

This is an interesting review that provides an important quantification of use of modified landscapes by primates, taking a global perspective. Detailed comments are provided below, but the key recommendations to improve this manuscript are:

1. Use a more sophisticated definition of matrix. It is confusing to refer to any modified land as a matrix, because when species live in that "matrix", then the matrix is also a habitat patch. This overly simplistic definition of matrix has led to at least one case of incorrect interpretation in this manuscript (see 116 below). A more useful definition takes a species-perspective, as explained by your reference no. 6. The extension of using a species-based definition of matrix is that you need to more clearly differentiate patch-dependent species from those that live in the modified habitat. This will lead to a more nuanced and informative perspective: we will then see how many species live in modified habitats, how many are still declining despite that, how many depend on forest and how many of those may derive supplementary benefits from elements of the modified landscape. This update requires substantial revision and careful thought to integrate to this manuscript, but it will result in a much clearer, less ambiguous story that is more enlightening.
2. Better highlight what is novel, which from my perspective is the quantification of effects and proportions that use different landscape elements to different extents, rather than the observation that some primates use modified habitats.

16-17 this is a pretty standard conclusion. What's the new insight? Quantifying the species that benefit or conversely, quantifying the species that don't? Or identifying how to modify the matrix so that more species benefit?

20-28. This is a confusing use of 'matrix'. If a species lives in the fields then it is a habitat patch to that species. To avoid having to explain that matrix can also be habitat and therefore matrix has no functional meaning from a conservation perspective, it is better to use the definition of matrix that includes the species' perspective. That is, the fields or other modified areas are only a matrix for species that do not have self sustaining populations in them. Thus you would refer to fields as a matrix when talking about species that do not have self sustaining populations in fields. Otherwise refer to fields (or other modified habitat) as fields (urban areas etc), without implying they are also matrix.

39. The improved definition of matrix is important because in this line you refer to species that are primarily in the original vegetation. These are patch-dependent species for which the farmland is a matrix. It's an important distinction to make. It is not very profound to say that species living in fields have lower extinction rates in landscapes consisting mostly of fields, because that is obvious. It is more interesting however in the context of patch dependent species. Defining how many primates live in fields is useful. But for those species dependant on original vegetation, the question of the extent of field use is more useful.

46. Is countryside biogeography a framework? Actually it's a wrapper term attempting to sweep up existing terms used in the habitat fragmentation literature.

98. Can you explain in the caption to fig 1b what the comparison with global number is?

116-118. If some species live in the modified landscape then they are not using it for supplementary purposes. Again, it is important to distinguish those species for which modified land is not a matrix. Without doing this, interpretation is imprecise at best, although in this example, interpretation is actually incorrect.

126. In fig 2 Could you add the number of species in each category? It would make a more useful take home statistic to be able to say 15 out of 45 species (for example) using corridors were foraging.

150-151. But if a species lives in villages or fields etc then its habitat is not fragmented. It is important to distinguish patch dependent species from those that live in modified areas. This should be easy to do as you know which species undertake all activities in modified habitats. Essentially what you currently do is confound the patch dependent and disturbed-area species. This is problematic because it is not very interesting to report that species that live in modified areas are increasing (though saying how many are in this category is useful, and if any are declining, that is also important). On the other hand if patch dependent species increase if they occasionally forage in open areas, there are supplementary benefits and this is a really interesting observation.

181-2. But Presumably these small species reach carrying capacity then individuals at the edges would have a need to head out of the forest. I don't think the "need" argument works.

182-3 this seems tautological. By definition if a species is a forest specialist it must occur more often in forest?

219-21. If this sentence sums up what you have done it is not breaking new ground.

Generally the concluding paragraph doesn't highlight what is new. Quantification of responses is new.

The focus on promoting the importance of the matrix seems a bit misplaced as 70% of species don't use it and are consequently more threatened. Reiterating how many species don't benefit is also a useful conclusion.

The concluding paragraph also claims adding resources into modified landscapes would be useful. It would be useful to have discussed evidence for what proportion of species would benefit from this and give some examples in the text earlier in the manuscript. Are the species that would benefit of conservation concern?

Reviewer #2 (Remarks to the Author):

This is a very interesting paper with well-supported data and convincing results. The statistical and phylogenetic analyses seem appropriate.

My main comment concerns the interpretation of the sensitivity analysis of the phylogenetic signal. This analysis seems to be a very important component to understand the influence of phylogeny on matrix tolerance; yet, the outputs are buried in Supplementary materials (within methods) and are not discussed well in the results. I think it would be really interesting to discuss these along with the Lepilemuridae results (line 208-216). It would also be great to see a discussion of what mechanism might explain why Lepilemuridae had such a big influence on the phylogenetic signal in matrix use.

Also, for the summary, the authors should talk about the results related to the phylogenetic analysis. The conclusion in the summary (line 16-17) could be made stronger by highlighting what practical conservation actions are recommended based on the overall findings.

Reviewer #3 (Remarks to the Author):

This is an original and timely study on the use of the anthropogenic matrix by non-human primates (sic). This is an Order level study, and the authors did an extraordinary job in compiling data from various sources. The paper is well written, databases look precise and exhaustive, and are per se a

great contribution. Analysis are surprisingly simple, but tests are correct and results are compelling. The conclusions are well supported by data.

Their findings represent an important basis for understanding the human-biodiversity interface, and in particular, set the stage for future research on the topic. I therefore believe that this is an important contribution for researchers working in diverse fields, from conservation biology, to behavioral ecology or human socioecology.

I really do not have any major suggestions or corrections, just a few minor comments, so I believe that the manuscript should be accepted.

First, authors could mention in the methods section what is their opinion about the reliability of the data sources that were used, and whether they believe that data heterogeneity could may have been a source of error for the analysis and interpretations. For instance, the majority of primate studies are short-term and based on a small number of individuals, but then there are some long-term, population-wide studies that may contribute very complete data. Although I have no problem with the chi-square tests that were used, there are more sophisticated ways to perform meta analysis, in which measurement errors may be incorporated, so the decision to use the chi-square tests could be explained.

Related with this, second, please note that it has been consistently demonstrated that time spent feeding is not a reliable measure of diet. Foods that require longer processing times (e.g. chewing) will lead to longer feeding times, although they may represent only a small amount of total food intake (e.g., leaf consumption by frugivore-folivore primates, such as howler monkeys). Could your negative results for trophic guild be a consequence of this?

L286: not clear to me why body mass is an ecological trait. I understand that it may be associated with ecological aspects, but it is an morphological trait, right?

L296-299: clarify that the expected values are number of species, as this is not currently evident.

I congratulate you for this excellent contribution and wish you every continued success in this promising line of research.

Pedro Dias

Universidad Veracruzana

Reviewer #4 (Remarks to the Author):

In this paper, the authors collate massive datasets to determine the global effect of human-modified landscapes on primates. By combining ecological, life history and phylogenetic information, alongside habitat classifications, the authors identify patterns that illustrate the characteristics of species that can persist with a “habitat-sharing” model of ecosystem conservation, versus those that require a “habitat-sparing” model.

Overall, I enjoyed reading this paper very much, and have very little to say by way of critique. The datasets are impressive, all analyses clearly justified and explained, and the results are placed in a contemporary literature. The insights are novel, and make a valuable contribution to our understanding of the effects of human activities on primates generally, and their conservation more specifically. I hope this paper will be widely shared, cited and used, so that conservation planners can use the results to help determine which conservation strategies will work best for their context/species.

One comment I had was whether the authors considered controlling for false discovery in their chi-squared analyses? By using the same control dataset, to compare against multiple test datasets, it is possible that multiple significant findings could occur by chance. I did not see any mention of alpha correction in the methods or main text.

This paper is very well written, in fact I only found two possible typos: L284 delete “and” before “body”; L490 spelling: “advice”.

Response to reviewers' comments

REVIEWER #1:

This is an interesting review that provides an important quantification of use of modified landscapes by primates, taking a global perspective. Detailed comments are provided below, but the key recommendations to improve this manuscript are:

1. Use a more sophisticated definition of matrix. It is confusing to refer to any modified land as a matrix, because when species live in that "matrix", then the matrix is also a habitat patch. This overly simplistic definition of matrix has led to at least one case of incorrect interpretation in this manuscript (see 116 below). A more useful definition takes a species-perspective, as explained by your reference no. 6. The extension of using a species-based definition of matrix is that you need to more clearly differentiate patch-dependent species from those that live in the modified habitat. This will lead to a more nuanced and informative perspective: we will then see how many species live in modified habitats, how many are still declining despite that, how many depend on forest and how many of those may derive supplementary benefits from elements of the modified landscape. This update requires substantial revision and careful thought to integrate to this manuscript, but it will result in a much clearer, less ambiguous story that is more enlightening.

- We understand the reviewer's concern and agree that defining 'matrix' as all modified land covers can be too simplistic and misleading. Matrix is usually referred to as the 'non-habitat' in classical metapopulation and island biogeography models. However, in contrast to these models that are typically based on binary landscapes composed of habitat and non-habitat (matrix), novel theoretical approaches (e.g. countryside biogeography) recognize that anthropic covers may represent habitats of different qualities, thus suggesting that such binary classification is neither accurate nor correct (also see the proposal of 'functional landscapes' by Fahrig et al. 2011 – Ecology Letters). Furthermore, as there is no available information on habitat use and preferences by the 147 primate species assessed in our review, we cannot do a binary classification of habitat and non-habitat. Therefore, to avoid confusion while remaining congruent with our research question and study design, we have changed the term 'matrix' to 'anthropic land covers', throughout the manuscript.
- Using this concept (i.e. anthropic land cover) we can assess "how many species live in modified habitats, how many are still declining despite that, how many depend on forest and how many of those may derive supplementary benefits from elements of the modified landscape" (see Fig. 4). To this end, we classified primates into two groups: forest specialists (i.e. patch-dependent) and non-forest specialist species. Results from such classification are in Fig. 4d, in which we show that forest specialist species seem to avoid human settlements and open areas.
- The studies we compiled reporting primates using anthropic land covers do not report if species are using these land covers as habitat (self-sustaining populations). Thus, we can only suggest that primates may be using these covers as temporal or even permanent habitat. We therefore suggest that "An important

next step will be to assess which species can maintain their populations solely in ALCs, which species are strongly dependent on their natural habitats, and which ones may survive in natural habitat patches with some degree of landscape supplementation in ALCs.” (lines 130-133).

2. Better highlight what is novel, which from my perspective is the quantification of effects and proportions that use different landscape elements to different extents, rather than the observation that some primates use modified habitats.

- To address this comment we have modified the main text as follows (please see lines 81-85): “While primates are known to use some types of ALCs³¹⁻³³, the available evidence is widely scattered and the global patterns of use remain unknown beyond a qualitative level. Further, no comprehensive effort exists to link primates’ ecological traits to the extent of use of specific ALCs, greatly limiting our ability to predict the impact of specific landscape-management strategies on these mammals.”. In particular, among other novel findings, we show that species using anthropic land covers are less often threatened with extinction, but more often diurnal, medium or large-bodied, not strictly arboreal, and habitat generalists. We also highlight these novel contributions in the abstract, and in the last paragraph of the main text (lines 26-28 and 211-217).

16-17 this is a pretty standard conclusion. What's the new insight? Quantifying the species that benefit or conversely, quantifying the species that don't? Or identifying how to modify the matrix so that more species benefit?

- Unfortunately the abstract can only contain 150 words, and we could not add text here. However, we have rewritten our last sentence as follows: “These novel findings provide valuable quantitative information for improving management practices for primate conservation worldwide.” (lines 34-35).

20-28. This is a confusing use of ‘matrix’. If a species lives in the fields then it is a habitat patch to that species. To avoid having to explain that matrix can also be habitat and therefore matrix has no functional meaning from a conservation perspective, it is better to use the definition of matrix that includes the species' perspective. That is, the fields or other modified areas are only a matrix for species that do not have self sustaining populations in them. Thus you would refer to fields as a matrix when talking about species that do not have self sustaining populations in fields. Otherwise refer to fields (or other modified habitat) as fields (urban areas etc), without implying they are also matrix.

- As explained above, we have changed the term ‘matrix’ to anthropic land covers to avoid confusion.

39. The improved definition of matrix is important because in this line you refer to species that are primarily in the original vegetation. These are patch-dependent species for which the farmland is a matrix. It's an important distinction to make. It is not very profound to say that species living in fields have lower extinction rates in landscapes consisting mostly of fields, because that is obvious. It is more interesting however in the context of patch dependent species. Defining how many

primates live in fields is useful. But for those species dependant on original vegetation, the question of the extent of field use is more useful.

- We agree with the reviewer, and thus, in addition to showing the global number of primate species that are able to use anthropic land covers, we show the number (and proportion) of forest dependent and non-forest dependent species that used each land cover type (Fig. 4d). This graph shows that forest specialist species are able to use all anthropic land covers, including human settlements and open areas.

46. Is countryside biogeography a framework? Actually it's a wrapper term attempting to sweep up existing terms used in the habitat fragmentation literature.

- Following Daily (1997 - cited in the text), and other studies (e.g. Wolfe et al. 2015 - Ecosphere), countryside biogeography can be defined as an ecological framework. Mendenhall et al. 2013 (Encyclopedia of Biodiversity, Volume 2, pp. 347-360) review the concept, and describe that it involves theoretical models, concepts and methods from different ecological disciplines, such as island biogeography, conservation biogeography and vicariance biogeography. Mendenhall et al. (2013) define countryside biogeography as the “study of the distribution of biological variation over space and time in human-dominated ecosystems” and describe how it is related not only to fragmentation research, but to other disciplines, such as conservation biology, ecosystem ecology, and evolution. In any case, it has been an emerging ecological approach to the study of biodiversity in human-modified landscapes, and thus, we changed the words ‘ecological framework’ to ‘ecological approach’ to avoid confusion (line 62).

98. Can you explain in the caption to fig 1b what the comparison with global number is?

- We have rewritten this caption as follows: “Proportion of species using **ALCs** (n = 147 species) compared to the total proportion of species (n = 504 species) in each biogeographic realm” (lines 493-495).

116-118. If some species live in the modified landscape then they are not using it for supplementary purposes. Again, it is important to distinguish those species for which modified land is not a matrix. Without doing this, interpretation is imprecise at best, although in this example, interpretation is actually incorrect.

- As stated above, we do not have enough information to identify which species are living permanently in anthropic land covers. To clarify this point, we have included in the methods section the following text: “As most studies did not report extended information about the use of each ALC, we cannot know if primates are using it as habitat.” (lines 267-268). We can only suggest that most species probably supplement their habitat in these emerging land covers. For example, we now clarify in lines 123-128 and 130-133 that: “Although most studies did not report if the species were using ALCs as habitat, at least 86 species (17% of all primates on Earth) are actively obtaining food resources from ALCs, highlighting their importance for primate conservation^{32,37}. In the case of forest-specialist primates, which represent 70% of the studied species, these results suggest that they can supplement their diet by foraging in ALCs – a

process referred to as “landscape supplementation” [...] An important next step will be to assess which species can maintain their populations solely in ALCs, which species are strongly dependent on their natural habitats, and which ones may survive in natural habitat patches with some degree of landscape supplementation in ALCs”.

126. In fig 2 Could you add the number of species in each category? It would make a more useful take home statistic to be able to say 15 out of 45 species (for example) using corridors were foraging.

- Figure 2 shows number of records, not species. Nonetheless, following the reviewer’s suggestion, we included the number of records in each anthropic land cover. With this information added it will be easier to interpret the extent of use of each land cover type (e.g. 20 of 52 records show primates foraging in human settlements).

150-151. But if a species lives in villages or fields etc then its habitat is not fragmented. It is important to distinguish patch dependent species from those that live in modified areas. This should be easy to do as you know which species undertake all activities in modified habitats. Essentially what you currently do is confound the patch dependent and disturbed-area species. This is problematic because it is not very interesting to report that species that live in modified areas are increasing (though saying how many are in this category is useful, and if any are declining, that is also important). On the other hand if patch dependent species increase if they occasionally forage in open areas, there are supplementary benefits and this is a really interesting observation.

- Again, it can be confusing to use the matrix concept, so we decided not to use it anymore, and use ‘anthropic land cover’ instead.

181-2. But Presumably these small species reach carrying capacity then individuals at the edges would have a need to head out of the forest. I don't think the “need” argument works.

- We agree with the reviewer and have modified the sentence making the point that it is a matter of time, i.e. small primates with small home ranges are able to persist in habitat remnants for longer time periods, thus lowering the probability of recording them in anthropic land covers in landscapes with a relatively recent land-use change process (lines 176-179).

182-3 this seems tautological. By definition if a species is a forest specialist it must occur more often in forest?

- To clarify our point, we now state that “Even though forest-specialists were less frequent among ALC-tolerant species ($\chi^2 = 11.19, P = 0.003$), particularly in human settlements ($\chi^2 = 31.53, P < 0.001$) and open areas ($\chi^2 = 11.76, P = 0.003$) (Fig. 4d), they were present in all land cover types.” (lines 179-182).

219-21. If this sentence sums up what you have done it is not breaking new ground.

- We have rewritten this sentence and, in general, modified the whole paragraph to strengthen our conclusions and highlight our contributions (lines 209-229).

Generally the concluding paragraph doesn't highlight what is new. Quantification of responses is new.

- Thank you for your suggestion. As we commented above, we have strengthened our concluding paragraph (please see lines 209-229).

The focus on promoting the importance of the matrix seems a bit misplaced as 70% of species don't use it and are consequently more threatened. Reiterating how many species don't benefit is also a useful conclusion.

- Great point! We have included the following text: “We note, however, that for 70% of the primates on Earth, we found no evidence of ALC use, suggesting that benefits associated with ALCs are limited to some species, in which case they are unlikely to prevent the extinction crisis of the world’s primates” (line 217-220).

The concluding paragraph also claims adding resources into modified landscapes would be useful. It would be useful to have discussed evidence for what proportion of species would benefit from this and give some examples in the text earlier in the manuscript. Are the species that would benefit of conservation concern?

- As the article summarizes evidence of primate species that use anthropic land covers, that is one of our outcomes. We argue that all (n = 147 species) primates that are able to use anthropic land covers can, to some extent, take advantage from human-modified landscapes by foraging, resting or travelling in these covers (Fig. 2). Moreover, Figure 3 shows the conservation status of the species that are benefiting. As we have seen, differences in habitat requirement influence the use of anthropic land covers, such that habitat generalist species are more likely to use open areas and human settlements than expected, whereas arboreal and/or forest specialist species are more likely to be restricted to using land covers structurally similar to primary forest, such as secondary vegetation, connectors and arboreal crops.

REVIEWER #2

This is a very interesting paper with well-supported data and convincing results. The statistical and phylogenetic analyses seem appropriate.

My main comment concerns the interpretation of the sensitivity analysis of the phylogenetic signal. This analysis seems to be a very important component to understand the influence of phylogeny on matrix tolerance; yet, the outputs are buried in Supplementary materials (within methods) and are not discussed well in the results. I think it would be really interesting to discuss these along with the Lepilemuridae results (line 208-216). It would also be great to see a discussion of what mechanism might explain why Lepilemuridae had such a big influence on the phylogenetic signal in matrix use.

- We moved key points from the supplementary material to the main text, in a new paragraph summarizing the sensitivity analysis and our interpretation. We now discuss how we expected Cercopithecidae to have a strong influence on the overall phylogenetic pattern because of its rich species diversity, but instead found that Lepilemuridae was the only clade to influence the phylogenetic signal values significantly when excluding it from the analyses. We related this result to the ecological traits shared by all species of Lepilemur, such as being strictly arboreal, nocturnal, and forest-dependent (lines 195-207).

Also, for the summary, the authors should talk about the results related to the phylogenetic analysis.

- The abstract is limited to 150 words, but we have added a short sentence on these results (line 31-32).

The conclusion in the summary (line 16-17) could be made stronger by highlighting what practical conservation actions are recommended based on the overall findings.

- We have changed the last sentence of the abstract as follows: “These novel findings provide valuable quantitative information for improving management practices for primate conservation worldwide” (line 34-35). Unfortunately, space limitation (150 words) does not allow us to include more details.

REVIEWER #3

This is an original and timely study on the use of the anthropogenic matrix by non-human primates (sic). This is an Order level study, and the authors did an extraordinary job in compiling data from various sources. The paper is well written, databases look precise and exhaustive, and are per se a great contribution. Analyses are surprisingly simple, but tests are correct and results are compelling. The conclusions are well supported by data.

Their findings represent an important basis for understanding the human-biodiversity interface, and in particular, set the stage for future research on the topic. I therefore believe that this is an important contribution for researchers working in diverse fields, from conservation biology, to behavioral ecology or human socioecology.

I really do not have any major suggestions or corrections, just a few minor comments, so I believe that the manuscript should be accepted.

First, authors could mention in the methods section what is their opinion about the reliability of the data sources that were used, and whether they believe that data heterogeneity could have been a source of error for the analysis and interpretations. For instance, the majority of primate studies are short-term and based on a small number of individuals, but then there are some long-term, population-wide studies that may contribute very complete data.

- Thank you for your comment. We are not sure if the reviewer refers to the studies reviewed to quantify the use of anthropic land covers (ALC; following a comment by reviewer 1 we have replaced ‘matrix’ by ‘anthropic land cover throughout the manuscript) by primates (n = 258 studies), or to the studies reviewed to obtain the information on the ecological traits of primates (n = 370 studies). In the former case, we do not think that there can be any bias because we selected the most recent 60 studies for each ALC, and the use of ALC was assessed on a presence/absence basis (e.g. number of species that are reported to use human-settlements). Thus, we do not think that differences in sample size or methodology among studies can have a major bias in our findings, as a single observation of primates using a given ALC type is evidence of use.
- Yet, for the latter case, we now clarify in the methods that we are confident on the reliability of the data sources, not only because we focused on studies published in peer-reviewed scientific journals and books, but also because we checked all functional traits in sources produced by experts, such as Mittermeier et al. (2013) and All the World’s Primates’ project (<https://www.alltheworldsprimates.org>). We also checked carefully the database for possible errors, and we excluded non-reliable records (e.g., very extreme or contradictory values, and values obtained with questionable methodology). Furthermore, to avoid any bias, we focused on categorical traits because continuous traits are probably more dependent on sample size and methodology. In particular, we considered: locomotion mode (i.e., arboreal, terrestrial or both), diel activity (i.e., diurnal, nocturnal or cathemeral), body mass and trophic guild. Locomotion mode and diel activity are relatively well-known, and are not affected by sample size or methodology. Body size can depend on sample size, but to reduce any bias, this variable was categorized into three classes: small (<2 kg), medium (2-10 kg), and large (>10 kg). Probably, the ecological trait more sensitive to sample size and methodology can be the trophic guild, so this variable was classified into six general groups: frugivorous (>60% of fruits in diet), folivorous (>60% leaves in diet), folivore-frugivorous (diet comprised of both fruits and leaves in similar proportions), omnivorous (both plants and animals in diet), insectivorous (diet dominated by arthropods) and gummivorous (diet dominated by plant exudates).

Although I have no problem with the chi-square tests that were used, there are more sophisticated ways to perform meta analysis, in which measurement errors may be incorporated, so the decision to use the chi-square tests could be explained.

- We did not use meta-analysis because this methodological approach works with ‘effect sizes’ (e.g. slope in a linear regression) compiled from different studies. This information is not available because the use of anthropic land covers by primates is rarely quantified in most primate studies, and when observed, records are mostly anecdotal (e.g. ad libitum observations). We therefore based our study on frequency of occurrence of a given trait (e.g. number of species with large body size). To assess if there is a functional signal in the use of anthropic land covers, we used chi-square goodness-of-fit tests because this analytical approach is adequate to determine whether observed sample frequencies (e.g. number of diurnal species among species using ALCs) differ significantly from expected frequencies. In our case, the expected frequencies are based on all the world’s primates. For example, if 25% of primates on Earth

are nocturnal, our null hypothesis is that 25% of ALC-tolerant species should be nocturnal. Significant deviations from this expected value indicate that there is a functional signal (e.g. in average, only 10% of ALC-tolerant species were nocturnal, suggesting that nocturnality imposes certain limitations that prevent ALC use). We justify the use of Chi tests in lines 300-302.

Related with this, second, please note that it has been consistently demonstrated that time spent feeding is not a reliable measure of diet. Foods that require longer processing times (e.g. chewing) will lead to longer feeding times, although they may represent only a small amount of total food intake (e.g., leaf consumption by frugivore-folivore primates, such as howler monkeys). Could your negative results for trophic guild be a consequence of this?

- This is an interesting point. Nevertheless, we do not see how this shortcoming could bias our results in favor of non-significant results because, for each species, trophic guild was obtained from several sources. In particular, this ecological trait was extracted primarily from Mittermeier et al. (2013) – a massive encyclopedia that compiles information from several studies, and thus, it includes studies with different sample sizes and methodologies. To further reduce such potential bias, trophic guild was assessed as a categorical variable, including six very general groups: frugivorous (>60% of fruits in diet), folivorous (>60% leaves in diet), folivore-frugivorous (diet comprised of both fruits and leaves in similar proportions), omnivorous (both plants and animals in diet), insectivorous (diet dominated by arthropods) and gummivorous (diet dominated by plant exudates).

L286: not clear to me why body mass is an ecological trait. I understand that it may be associated with ecological aspects, but it is a morphological trait, right?

- Yes, body mass is a morphological trait, but of high ecological significance. For instance, body mass is related to home range, thus affecting the way species interact with their environment. Body mass also has significant impacts across multiple scales of biological organization, from individuals to ecosystems (Peters and Peters 1986; Gaston and Blackburn 2000; Savage et al. 2004; Woodward et al. 2005). Therefore, we clarified in lines 280-283, that although body mass is a morphological trait, it can be considered an ecological trait.

L296-299: clarify that the expected values are number of species, as this is not currently evident.

- Done! We have included “In particular, we compared the number of species with each trait between those species that were observed using ALCs and the expected values based on all of the world’s primates” in line 300-302.

I congratulate you for this excellent contribution and wish you every continued success in this promising line of research.

**Pedro Dias
Universidad Veracruzana**

REVIEWER #4

In this paper, the authors collate massive datasets to determine the global effect of human-modified landscapes on primates. By combining ecological, life history and phylogenetic information, alongside habitat classifications, the authors identify patterns that illustrate the characteristics of species that can persist with a “habitat-sharing” model of ecosystem conservation, versus those that require a “habitat-sparing” model.

Overall, I enjoyed reading this paper very much, and have very little to say by way of critique. The datasets are impressive, all analyses clearly justified and explained, and the results are placed in a contemporary literature. The insights are novel, and make a valuable contribution to our understanding of the effects of human activities on primates generally, and their conservation more specifically. I hope this paper will be widely shared, cited and used, so that conservation planners can use the results to help determine which conservation strategies will work best for their context/species.

One comment I had was whether the authors considered controlling for false discovery in their chi-squared analyses? By using the same control dataset, to compare against multiple test datasets, it is possible that multiple significant findings could occur by chance. I did not see any mention of alpha correction in the methods or main text.

- We understand the reviewers’ concern, but we did not find it necessary to apply alpha correction (e.g. Bonferroni) in our analyses because: (1) our P values were all small enough (i.e. most of them were <0.001 ; see Fig. 3 and 4), and (2) this kind of approach has been criticized because although they contribute to avoid Type I statistical error, they can increase the likelihood of Type II error.

This paper is very well written, in fact I only found two possible typos: L284 delete “and” before “body”; L490 spelling: “advice”.

- Thank you very much for your corrections! They were both corrected.

Reviewer #1 (Remarks to the Author):

The authors have addressed my previous concerns satisfactorily. I have only a few additional minor suggestions.

128-30. the relevant statistic here is the number of forest specialists that also use ALCs; how many of the 86 species that used ALCs are forest specialists?

153-55. Does this sentence just repeat the first sentence but mentioning the reverse of the same statistic? In fact, this paragraph seems to just say the same thing throughout but in slightly modified form. It would be better to start with the key observation (about human settlements and open areas), then develop that idea rather than just repeating it in different ways.

180-182. Is there any signal that the most threatened species are also larger bodied? Are forest-specialist large-bodied species more common among higher threat categories? This would at least be one way to evaluate your suggestion that smaller species are less vulnerable to extinction in small remnants.

164-189. In general this paragraph could grapple more deeply with trying to explain the observed patterns. Can you offer some plausible explanations for why nocturnal species are not in human settlements, as you expected? Also, literature on traits often recognizes trait syndromes or collection of traits that tend to go together and help explain responses to environmental variation. This might be a useful literature to help put your findings into a broader context. The paragraph is already very long; it would be better divided into shorter, more easily digested chunks.

201. say which family again to make it clear.

Reviewer #3 (Remarks to the Author):

In this revised version of the manuscript the authors answered properly to the comments I made. Their answers to other reviewers' comments and suggestions also seem correct to me. I therefore have no further revisions.

Just for clarification:

About my comment on data quality, 1) I was referring to arguments of ignorance, as you cannot ascertain that absence of evidence that a particular taxon is using ALC is evidence that it does not use it; 2) it may be more likely to observe ALC use after 1,000 observation hours than after 10.

About my comment on the use of time spent feeding issue, the idea is that 1) time spent feeding is a measure that biases diet classifications in favor of foods harder to process; 2) for many taxa, this is the measure that people consistently use; 3) if several species are being misclassified in terms of their trophic guild, your results could change.

Reviewer #4 (Remarks to the Author):

In this paper, the authors collate datasets to determine the global effect of human-modified landscapes on primates. This is a paper that I reviewed previously. In my previous review, I had a positive assessment of the work, and maintain that opinion. I have read the paper and found it to be improved from the previous version. In regards specifically to my previous comments, I accept the authors response. I hope that this work will be used to help make conservation decisions to

improve the status of primate species around the world.

REVIEWERS' COMMENTS

Reviewer #1 (Remarks to the Author):

The authors have addressed my previous concerns satisfactorily. I have only a few additional minor suggestions.

128-30. the relevant statistic here is the number of forest specialists that also use ALCs; how many of the 86 species that used ALCs are forest specialists?

- 70% of primate species using ALCs for feeding are forest specialists. We included this information in line 125.

153-55. Does this sentence just repeat the first sentence but mentioning the reverse of the same statistic? In fact, this paragraph seems to just say the same thing throughout but in slightly modified form. It would be better to start with the key observation (about human settlements and open areas), then develop that idea rather than just repeating it in different ways.

- Done. We removed the first sentence to avoid redundancy and start with the key observation (lines 147-153).

180-182. Is there any signal that the most threatened species are also larger bodied? Are forest-specialist large-bodied species more common among higher threat categories? This would at least be one way to evaluate your suggestion that smaller species are less vulnerable to extinction in small remnants.

- We modified this sentence to avoid confusion. We propose that the lower proportion of small-bodied species in ALCs may be related to two important factors: (1) these species are more likely to be arboreal, which limits their movement into treeless areas; and (2) small primates tend to have smaller home ranges, and thus, they can be able to inhabit smaller habitat remnants without using resources from ALCs. These two factors can lower the probability of observing small-bodied primates in ALCs (lines 168-173).
- Yet, we do not know if smaller species are less vulnerable to extinction in small remnants. We now clarified that: “Although this finding does not mean that small primates have lower extinction risk in human-modified landscapes, our results point in this direction, as 57.3% of large ALC-tolerant species (n = 21 species) are threatened with extinction, whereas 49.9% of small-bodied species (n = 42) are threatened (Supplementary Table 1). In contrast, other studies suggest that those species that avoid using ALCs can be more prone to extinction in these emerging landscapes^{10,47,48}; thus, additional primate studies are needed to accurately assess the effect of body weight on extinction risk” (lines 173-180).

164-189. In general this paragraph could grapple more deeply with trying to explain the observed patterns. Can you offer some plausible explanations for why nocturnal species are not in human settlements, as you expected?

- Yes, as stated in lines 158-161, “We would have expected a higher (not lower) incidence of nocturnality among species using ALCs because nocturnal primates are less likely to encounter humans, and thus, they could perceive ALCs as less dangerous, compared to diurnal primates”. Then, why nocturnal species seem to avoid the use of ALCs? We propose that “nocturnal primates are all forest specialists, arboreal and with small-to-medium body mass – ecological traits that together seem to limit the use of ALCs” (lines 161-163).

Also, literature on traits often recognizes trait syndromes or collection of traits that tend to go together and help explain responses to environmental variation. This might be a useful literature to help put your findings into a broader context.

- Great point! We added four papers on species traits that can predict species sensitivity to habitat disturbance in human-modified landscapes (Milton & May 1976 – Nature; McKinney & Lockwood 1999 – Trends Ecol Evol; Boyle & Smith 2010 – Biol Conserv; Henle et al. 2004 – Biodivers Conserv).

The paragraph is already very long; it would be better divided into shorter, more easily digested chunks.

- Done. We broke it into three smaller paragraphs.

201. say which family again to make it clear.

- Done

Reviewer #3 (Remarks to the Author):

In this revised version of the manuscript the authors answered properly to the comments I made. Their answers to other reviewers’ comments and suggestions also seem correct to me. I therefore have no further revisions.

Just for clarification:

About my comment on data quality, 1) I was referring to arguments of ignorance, as you cannot ascertain that absence of evidence that a particular taxon is using ALC is evidence that it does not use it; 2) it may be more likely to observe ALC use after 1,000 observation hours than after 10.

About my comment on the use of time spent feeding issue, the idea is that 1) time spent feeding is a measure that biases diet classifications in favor of foods harder to process; 2) for many taxa, this is the measure that people consistently use; 3) if several species are being misclassified in terms of their trophic guild, your results could change.

- We are grateful for these clarifications.

Reviewer #4 (Remarks to the Author):

In this paper, the authors collate datasets to determine the global effect of human-

modified landscapes on primates. This is a paper that I reviewed previously. In my previous review, I had a positive assessment of the work, and maintain that opinion. I have read the paper and found it to be improved from the previous version. In regards specifically to my previous comments, I accept the authors response. I hope that this work will be used to help make conservation decisions to improve the status of primate species around the world.

- Thank you very much this positive feedback.